# A Simplified Spatial Methodology for Assessing Land Productivity Status in Africa

**Barasa Bernard \*, Majaliwa J. G. Mwanjalolo, Banduga Moses, Katwere James, Magaya Paul** **, Sadadi Ojoatre** **, Wanjiru Lydia and Margaret N. Walusimbi**

GEO-MIK Consultants Africa, Kampala Uganda, Plot 47, Nsamizi Road, Nsamizi View, Entebbe P.O. Box 36742, Uganda; majaliwam@gmail.com or majaliwa@geomikafrica.com (M.J.G.M.); moses@geomikafrica.com (B.M.); katxeus@gmail.com or katwere@geomikafrica.com (K.J.); magayait@gmail.com or paul@geomikafrica.com (M.P.); sadadi@geomikafrica.com (S.O.); lydia@geomikafrica.com (W.L.); walusimbi@geomikafrica.com (M.N.W.)
\* Correspondence: barasa@kyu.ac.ug; Tel.: +256-701-712-526

**Abstract:** The degradation of soil, vegetation and socio-economic transformations are a huge threat to Africa's land production. This study aimed to (i) assess the soil and land productivity of standing biomass and (ii) determine the effect of rainfall on the standing biomass in Eastern Africa. Soil productivity was determined using the Soil Productivity Index (SPI) and a simplified model was developed to estimate the Net Primary Productivity (NPP). The SPI indicators used included soil-organic matter, texture, soil moisture, base-saturation, pH, cation-exchange-capacity, soil-depth and drainage. The inputs of the simplified model are: MODIS Soil Adjusted Vegetation Index (SAVI), soil erosion, soil nutrient content and input, rainfall, land-use/cover and agro-ecological zones. The findings reveal that the countries with the most productive soils are Mauritius, Rwanda and South Sudan—while, for standing biomass, the countries with the highest spatial extent are Mauritius (97%), Rwanda (96%), Uganda (95%), South Sudan (89%), Ethiopia (47%) and Kenya (36%). Standing biomass is dominant in biomes such as natural forests, woodlands, croplands, grasslands, wetlands and tree-plantations. High land productivity was attributed to soil quality and management, land policy reforms, favourable climatic conditions and sustainable land husbandry activities. Rainfall was significantly correlated with standing biomass in most of the studied countries ($p < 0.05$) except Djibouti and Rwanda. Therefore, monitoring soil health, use and land reforms are key to sustaining vegetative biomass.

**Keywords:** land productivity; SAVI; soil erosion; QGIS; soil fertility; Africa



## 1. Introduction

In the past three decades, the demand for land by humans is steadily increasing worldwide. By this, more and more land is being converted for agricultural purposes at the expense of natural vegetation [1]. About 42% of the earth's human population is highly engaged in agriculture for survival throughout the year [2]. In Sub-Saharan Africa, agriculture employs between 60 and 80% of the population masses [3], and contributes over 36.4% of countries' GDP [4] and 60% of export earnings [5]. Despite this contribution, unsustainable agricultural activities are one of the major threats to soil health and associated land productivity. This is due to the intensification of unsustainable agricultural practices, vegetation degradation and mismanagement [6,7].

Land productivity refers to the capacity of a given soil to produce crop yield and support standing biomass in an ecosystem. In addition to human alteration, the state of land productivity is influenced by natural factors such as precipitation, soil chemical-biological-physical status, topography, the incidence of pests and diseases and land management [8] and the level of land degradation. The latter is the most pressing environmental problem that requires urgent attention in the global south because, if it is not attended to, it will

result in severe consequences such as food insecurity and chronic poverty. This has affected the world for centuries and is more pronounced in the developing world [9]. For example, in Ethiopia, the major causes of land degradation are rapid population increase, severe soil loss, deforestation and intensive crop cultivation [10]. As a result, the unending activities are responsible for soil nutrient deficit, extreme soil erosion, desertification, and conflicts [11,12], hence presenting a threat to vulnerable ecosystems and socio-economic development [13,14].

Therefore, frequent monitoring of land productivity is important for the sustenance of ecosystem services and goods. GIS and remote sensing tools are handy to monitor standing biomass [15,16]. The GIS tools can indirectly be utilised to evaluate land productivity such as through the development and integration of indices, mapping and interpolation [17]. The robust tools of GIS and earth observation can be effectively utilised to assess the potential of land productivity of any place in the world. It is also partly because they can ably monitor biomass on a large area of land at any given period of the year.

From our literature search, this study acknowledges that there is limited literature available related to land productivity assessments in Africa [18–20]. This is due to the value and importance attached to land for food production and hence the survival of most communities across the world, particularly in Sub-Saharan Africa [21–23]. This study, therefore, bridges this knowledge gap by providing a methodology for assessing land productivity at macro and micro levels, but also investigates productivity across a spectrum of different terrestrial biomes and agro-ecological zones and their causes.

The specific objectives of this research were to (i) assess the soil productivity and land productivity of standing biomass and (ii) determine the effect of rainfall on the standing biomass in Eastern Africa.

## 2. Materials and Methods

### 2.1. Study Area

The ten African countries selected to determine the state of land productivity are located in Eastern Africa. These include Djibouti, Eritrea, Ethiopia, Kenya, Mauritius, Rwanda, Somalia, South Sudan, Sudan and Uganda. They cover a landmass of about 224,006 km$^2$ (Figure 1). These countries were selected by the Global Monitoring for Environment Security (GMES) and Africa Programme. The objective of the GMES and Africa Programme is to address the growing needs of African countries to access and use Earth Observation (EO) data for the implementation of sustainable land development policies across the continent. This study was conducted between 2019 and 2020. The climate of studied countries can be broadly classified into arid to semi-arid (Horn of Africa), tropical (East Africa) and mild tropical maritime climate for Mauritius. The semi-arid areas experience a unimodal rainfall pattern while those in the tropical regions enjoy a bimodal pattern. On average, the studied countries receive total seasonal rainfall amounts of more than 500 mm which can support plant life. The study area's annual average temperature ranges from 21 °C to 31 °C. The coldest temperatures are recorded in Mauritius and the hottest in Khartoum [24]. The soils that support productivity are composed of Humic Latosols, Ferralsols, Acrisols, Nitosols, Lithosols, Vertisols, and Fluvisols [25,26]. Rudimentary smallholder subsistence farming is the main socio-economic activity in the region [27].

### 2.2. Datasets Collected and Used

The proposed methodology of land productivity was developed using freely available global earth observation datasets. However, this method can also be applied on customised data. The spatial datasets that were downloaded and used to assess land productivity included Globcover (2009), Shuttle Radar Topography Mission (SRTM) Digital Elevation Model (30 m), Food and Agriculture Organisation (FAO) soil data (2019), Climate Hazards center InfraRed Precipitation with Station data (CHIRPS) rainfall data (2001–2020), MODIS Soil Adjusted Vegetation Index (2001–2019) and land-use/cover (2015). The downloaded datasets were re-projected to WGS ESPG 4326 and collated with the national datasets

(Table 1). Each dataset was independently processed (such as inserting factor ratings) to meet the model requirements. The limitations of global spatial datasets used include coarse resolution and duplication of datasets. However, the global datasets were validated with country-specific information and they collated well.

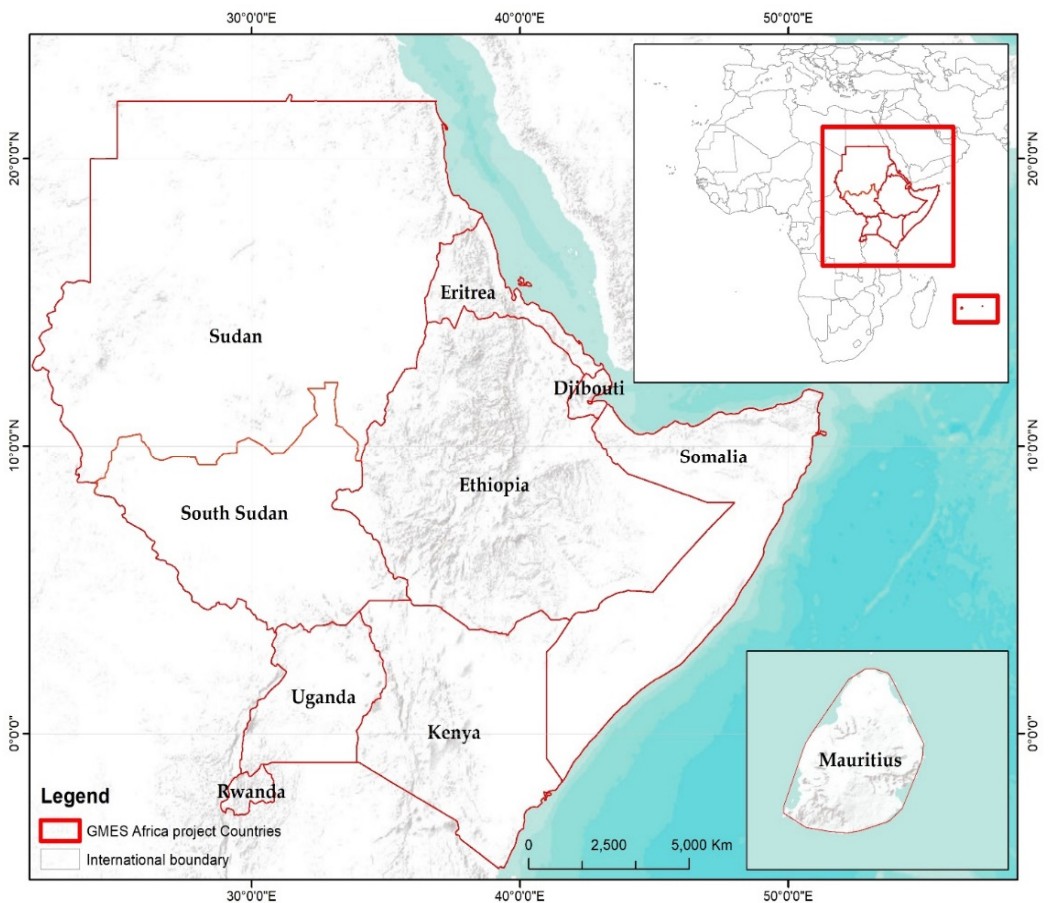

**Figure 1.** Location of studied countries.

**Table 1.** List of datasets and associated sources.

| Datasets | Source |
| --- | --- |
| Soils (with physio-chemical properties and FAO legend) | http://www.fao.org/soils-portal/ (accessed on 22 July 2020) |
| Climate data (rainfall) | https://data.giss.nasa.gov/impacts/agmipcf/agmerra/ (accessed on 24 July 2020) |
| Topography (Digital Elevation Model) | https://earthexplorer.usgs.gov/ (accessed on 4 August 2020) |
| Vegetation Indices (SAVI) | https://earthexplorer.usgs.gov/ (accessed on 6 June 2020) |
| Land use/cover | https://earthexplorer.usgs.gov/ (accessed on 2 July 2020) |
| Administrative boundaries | http://geoportal.rcmrd.org/ (accessed on 2 July 2020) |
| Protected areas | https://www.protectedplanet.net/ (accessed on 2 July 2020) |
| Population | National Bureau of Statistics (accessed on 4 August 2020) |
| Drainage | http://tapiquen-sig.jimdo.com (accessed on 2 July 2020) |
| Agro ecological zones | http://geoportal.rcmrd.org/layers/servir%3Aafrica_agroecological_zoning (accessed on 12 July 2020) |

*2.3. Development of Land Productivity Methodology*

2.3.1. Approach Used

An earth observation-based land productivity approach was adopted by this study. The approach proposed was based on the concepts of interacting human–environment systems [28], Millennium Ecosystem Assessment [29] and the Drivers–Pressures–State–Impacts–

Responses model [30,31]. These frameworks are preferred as the basis of reporting human–environment interactions to inform decision-making processes to curb land degradation.

This methodology takes into consideration recent efforts by Conservation International in designing the Trends.Earth tool [32]. Trends.Earth offers data and tools to inform the assessment of land degradation state, trend and performance (https://trends.earth/docs/en/index.html (accessed on 16 August 2020)). This platform uses cloud computing to process massive satellite images into usable information. It assesses land trends through three indicators: land productivity, land cover and soil carbon [33]. However, this approach does not factor in soil and social-economic information that have been recognized to influence land productivity, hence misreporting, for example, on seasonally used agricultural land, and harvested woodland plantations. However, the Trends.Earth tool recognises agro-ecological zones as a unit of state and performance comparison, and the determination of the relative productivity state is arbitrary. To bridge this gap, the proposed land productivity methodology intends to accurately define the above-ground productivity state and appropriate soil productivity. The definition of the above-ground productivity state is based on Mukuralinda et al. [34] productivity index—one of the robust statistical-based productivity indices. The latter assume that land productivity and degradation are the two extreme states of productivity. Any given state is a linear combination of the two extreme states, whose productivity index is given by the polarization index, similar to the NDVI, but where the parameters used in the polarisation index are the coefficients of the extreme states in the linear model used to define a given above ground productivity.

Therefore, the land productivity approach proposed considered the aspects of Soil Productivity and Net Primary Productivity-(NPP). Figure 2 shows the land productivity model. This model considers factors that influence ecosystem biomass production such as:

1.  Climate and climatic variables (rainfall, etc.);
2.  Ecosystem structural elements (like soil properties, slope, drainage, etc.);
3.  Vegetation health (trend);
4.  Human interactions (physical infrastructure, land use, demographic data, etc.).

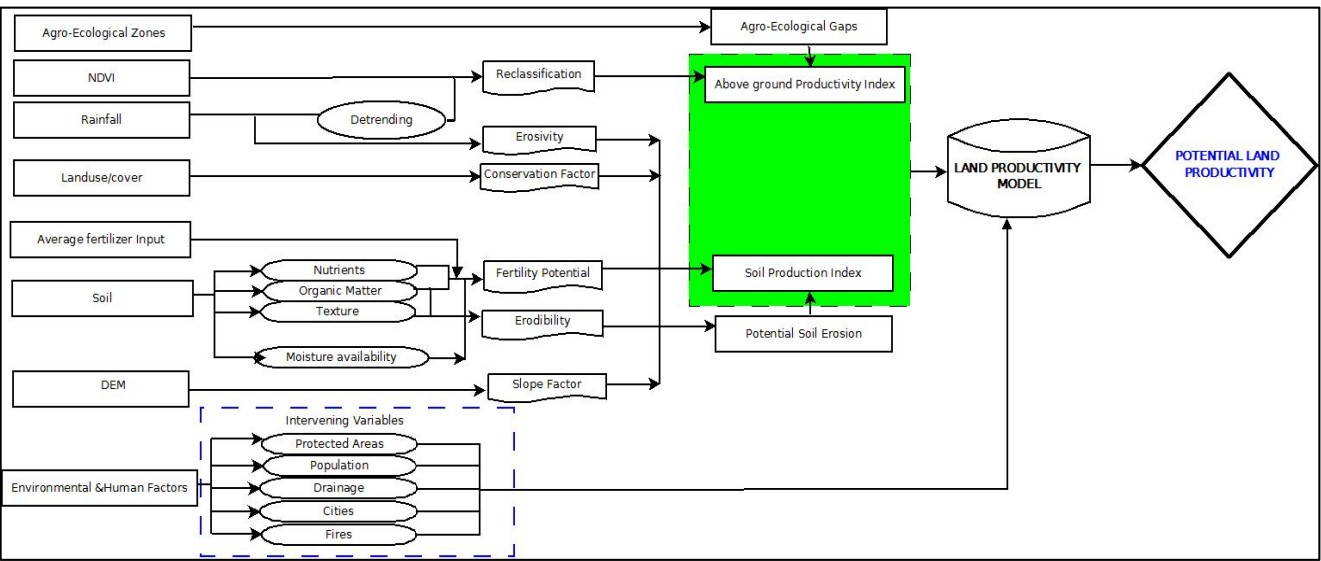

**Figure 2.** Land productivity model.

The Soil Productivity Index (SPI) indicators used in the model were soil organic matter, texture, moisture availability, gravel content, base saturation, pH, CEC, soil depth and drainage. However, the Net Primary Productivity (NPP) indicator used was the Soil-Adjusted Vegetation Index (SAVI). The overlay of soil and land productivity provides rich information on the overall potential of land productivity in a given area. A detailed description of processes followed to define SP and NPP are presented below.

### 2.3.2. Soil Productivity Index

The Soil Productivity Index was based on the Food and Agriculture Organisation (FAO) soil classification and Soil Productivity Index. The FAO soil classification provides inherent agricultural productivity of different soils, and infers them as low, moderate, high and very highly productive soils. This represents the potential of a given soil to produce crops or biomass. The inherent agricultural productivity of the soils of the targeted countries was based on FAO classification. However, soil fertility was not included in the methodology due to limited country specific data.

Factor ratings were defined for each range of diagnostic parameters used in the SPI and the different layers associated with these diagnostic parameters were reclassified to read the factor rating before the calculation of the SPI. Refer to Tables 2–10 for detailed factor ratings of soil depth, cation exchange, slope, soil texture, drainage, base saturation and pH, organic matter, soil moisture and slope depth. However, the soil gravel content did not significantly affect the soil productivity outcome; therefore, it was not considered in the final Soil Productivity Index. The soil productivity classes of very high and high were computed to estimate the most productivity coverage of soils while the intensities of very low and low were used to define countries with low productive soils.

The values of the SPI were standardized and reclassified based on the FAO factor rating scheme as shown below:

(a) Above 75—Very high productivity;
(b) <75—High productivity;
(c) <50—Moderate productivity;
(d) <30—Low productivity;
(e) <20—Very low productivity.

**Table 2.** Factor ratings of soil depth (cm).

| Soil Depth Classification | Codes | Class | Factor Rating |
|---|---|---|---|
| Very shallow | P1 | <10 | 20 |
| Shallow | P2 | 10–30 | 50 |
| Fairly deep | P3 | 30–90 | 70 |
| Deep | P4 | 90–120 | 85 |
| Very deep | P5 | >120 | 100 |

**Table 3.** Factor ratings of Cation Exchange Capacity (cmol·kg$^{-1}$).

| CEC | Code | Class | Factor Rating |
|---|---|---|---|
| Low | A0 | <5 | 20 |
| Moderate | A1 | <20 | 50 |
| Good | A2 | <40 | 75 |
| Very good | A3 | >40 | 100 |

**Table 4.** Factor ratings of slope (%).

| Slope | Code | Class | Factor Rating |
|---|---|---|---|
| Flat | E1 | 0–2' | 20 |
| Rolling | E2 | 2–8' | 50 |
| Moderately steep | E3 | 8–15' | 70 |
| Steep | E4 | 15–25' | 85 |
| Very steep | E5 | >25 | 100 |

**Table 5.** Factor ratings of soil texture.

| Texture | Code | Description | Factor Rating |
|---------|------|-------------|---------------|
| **Very low** | T1, T2 | Stony or gravel soils, extremely coarse textured soil | 20 |
| **Low** | T3, T5 | Dispersed clay of unstable structure or Heavy textured soils | 50 |
| **Moderate** | T6 | Medium heavy soils | 70 |
| **High** | T7 | Soils of averaged or balanced texture | 85 |
| **Very high** | T4 | Light textured soils | 100 |

**Table 6.** Factor ratings of drainage.

| Drainage | Code | Description | Factor Rating |
|----------|------|-------------|---------------|
| **Very poor** | D1 | Marked water logging all year round | 20 |
| **Poor** | D2 | Moderate water logging for 8 days to 2 months | 50 |
| **Moderate** | D3b | Water logging for brief period less than 8 days each time | 70 |
| **Good** | D3a | Good drainage-water table is sufficiently low | 85 |
| **Well** | D4 | Well drained soils | 100 |

**Table 7.** Factor ratings of base saturation and pH.

| Base Saturation and pH | Code | Class | Factor Rating |
|------------------------|------|-------|---------------|
| **Very low** | N1 | BS < 15%, pH:3.5–4.5 | 20 |
| **Low** | N2 | BS 15–35%, pH:4.5–5 | 50 |
| **High** | N3 | BS 35–50%, pH:5–6 | 85 |
| **Very high** | N4 | BS 50–75%, pH:6–7 | 100 |
| **Moderate** | N5 | BS > 75%, pH:7–8.5 | 70 |

**Table 8.** Factor ratings of organic matter (%).

| Organic Matter | Code | Description | Factor Rating |
|----------------|------|-------------|---------------|
| **Very low** | O1 | Very little organic matter < 1% | 20 |
| **Low** | O2 | Little organic matter 1–2% | 50 |
| **Moderate** | O3 | Average organic matter 2–5% | 70 |
| **High** | O4 | High organic matter >5% | 85 |
| **Very high** | O5 | Very High organic matter >5% and C/N over 25 | 100 |

**Table 9.** Factor ratings of soil moisture content (%).

| Code | Description and Classes | Factor Rating |
|------|-------------------------|---------------|
| **O1** | Very little organic matter, less than 1% | 20 |
| **O2** | Little organic matter, 1–2% | 50 |
| **O3** | Average organic matter content, 2–5% | 70 |
| **O4** | High organic matter content, over 5% | 85 |
| **O5** | Very high content but C/N over 25 | 100 |

**Table 10.** Factor ratings of topography (slope depth—cm).

| | Soil Depth (P) | |
|---|---|---|
| Code | Description | Factor Ratings |
| **P1** | Rock outcrops with no soil cover or very shallow cover | 20 |
| **P2** | Very shallow soil, <30 cm | 30 |
| **P3** | Shallow soil, 30–60 cm | 50 |
| **P4** | Fairly deep soil, 60–90 cm | 70 |
| **P5** | Deep soil 90–120 cm | 85 |
| **P6** | Very deep soil > 120 cm | 100 |

2.3.3. Estimation of Soil Erosion

Soil loss was estimated based on the Revised Universal Soil Loss Equation (RUSLE). Only the potential soil loss was determined (Management factor P = 1). This process is important to understand reductions in land productivity. Soil loss was computed based on erosivity, erodibility, slope length and cover factor. Rainfall erosivity was determined based on Moore [35]:

$$= 0.029 \times (3.96 \times \text{P} + 3122) - 26 \tag{1}$$

- where P is the mean annual precipitation in mm. P was determined for 30 years using CHIRPS rainfall data.
- Various approaches can be used to estimate soil erodibility. In this method, erodibility was estimated based on Morgan [36].
- Slope length was estimated based on Moore et al. [37]. This is expressed in the equation as:

$$LS = \left( Flow\ length(or\ Flow\ Accumulation) \times \frac{Cellsize}{22.13} \right)^{0.4} \times \left( \frac{sinslope}{0.0896} \right)^{1.3}. \tag{2}$$

- Cover factors were extracted from Panagos et al. [38] as summarised below in Table 11:

**Table 11.** C cover factors for Soil Erosion Assessment.

| No | Land Use/Cover Types | C Factor |
|---|---|---|
| 1 | Trees cover areas | 0.13 |
| 2 | Shrubs cover areas | 0.3 |
| 3 | Grassland | 0.3 |
| 4 | Cropland | 0.5 |
| 5 | Vegetation aquatic or regularly flooded | 0 |
| 6 | Lichen Mosses/Sparse vegetation | 0.45 |
| 7 | Bare areas | 0.9 |
| 8 | Built-up areas | 0.9 |
| 10 | Open water | 0 |

Soil loss was reclassified using FAO [39] classification scheme as presented below:

a. 0–2 ton/ha/yr—Very low
b. 2–10 ton/ha/yr—Low
c. 10–50 ton/ha/yr—Moderate
d. 50–90 ton/ha/yr—High
e. Above 90 ton/ha/yr—Very high

### 2.3.4. Net Productivity Index

Net Primary Productivity (NPP)/standing biomass is one of the most important parameters in describing the functioning of any ecosystem [40–42]. The MODIS (MOD13Q1) Soil Adjusted Vegetation Index (SAVI) (2001–2019) data were downloaded and used to determine the country-specific status of land productivity. MODIS (SAVI) and CHIRPS (rainfall data) were used to investigate if rainfall had a significant impact on standing biomass.

### 2.3.5. Land Productivity Classification Scheme

The classification scheme of land productivity was developed in consideration of ago-ecological zones of the studied countries where the standing biomass value range thresholds were defined using eight percentiles and aggregated into a realistic land productivity scheme that is acceptable. To compute the spatial extent of land productivity status, the intensity classes of very high, high and slightly high were considered while extremely low, very low and low classes were used to establish countries with the least productivity of standing biomass.

### *2.4. Effect of Rainfall on Land Productivity*

Reference monitoring points were generated in each agroecological zone in the studied countries and used to assess the relationship between rainfall and Soil Adjusted Vegetation Index using regression models between the 2001 and 2019 periods. Depending on the number of agroecological zones in each country, the monitoring points were Kenya (7), Uganda (8), Rwanda (3), Sudan (9), South Sudan (2), Somalia (6), Ethiopia (10,) Eritrea (2) and Djibouti (2).

### *2.5. Validation of Land Productivity Maps*

To confirm the accuracy of the developed maps, the mandated national institutional stakeholders in partnership with the GMES Africa programme through the Regional Centre for Mapping of Resources for Development (RCMRD) were selected and consulted. These included members of the public, private sector and community-based organisations. Project consultations were virtually conducted at the peak of the COVID-19 pandemic, using a validated interview guide and responses recorded. The guide thought to investigate the status of land productivity, indicators, causes and effects. The Zoom video communication platform was used to interview the respondents primarily because it is secure and reliable. In all the studied countries, a second in-country validation of the products was conducted through the presentation of findings in a face-to-face nationally organised workshop upon easing the pandemic lockdowns. The national key informants were formally invited for a day's validation workshop and further provided feedback on the land productivity outputs.

### 3. Results

### *3.1. Soil Productivity Index*

This study reveals that the SPI in the studied countries was dominant from "Very Low" to "Low" (74.2%). In other words, the majority of these soils fall under low and very low Soil Productivity Indices. Those with "High" to "Very High" productivity indices only represent 11.8% of the studied countries (Table 12 and Figure 3). At the country level, the most productive soils are found in Mauritius (79%), Rwanda (40%) and South Sudan (19%) in terms of spatial extent—whereas, the countries with the least productive soils are Djibouti (96%), Eritrea (92%), Kenya (84%) and Sudan (80%) (Table 13).

**Table 12.** Regional-scale of soil productivity for the studied countries.

| SPI | Area (km$^2$) | % |
|---|---|---|
| Very Low | 1,604,887 | 30.2 |
| Low | 2,338,109 | 44.0 |
| Moderate | 750,347 | 14.1 |
| High | 436,166 | 8.2 |
| Very High | 189,527 | 3.6 |

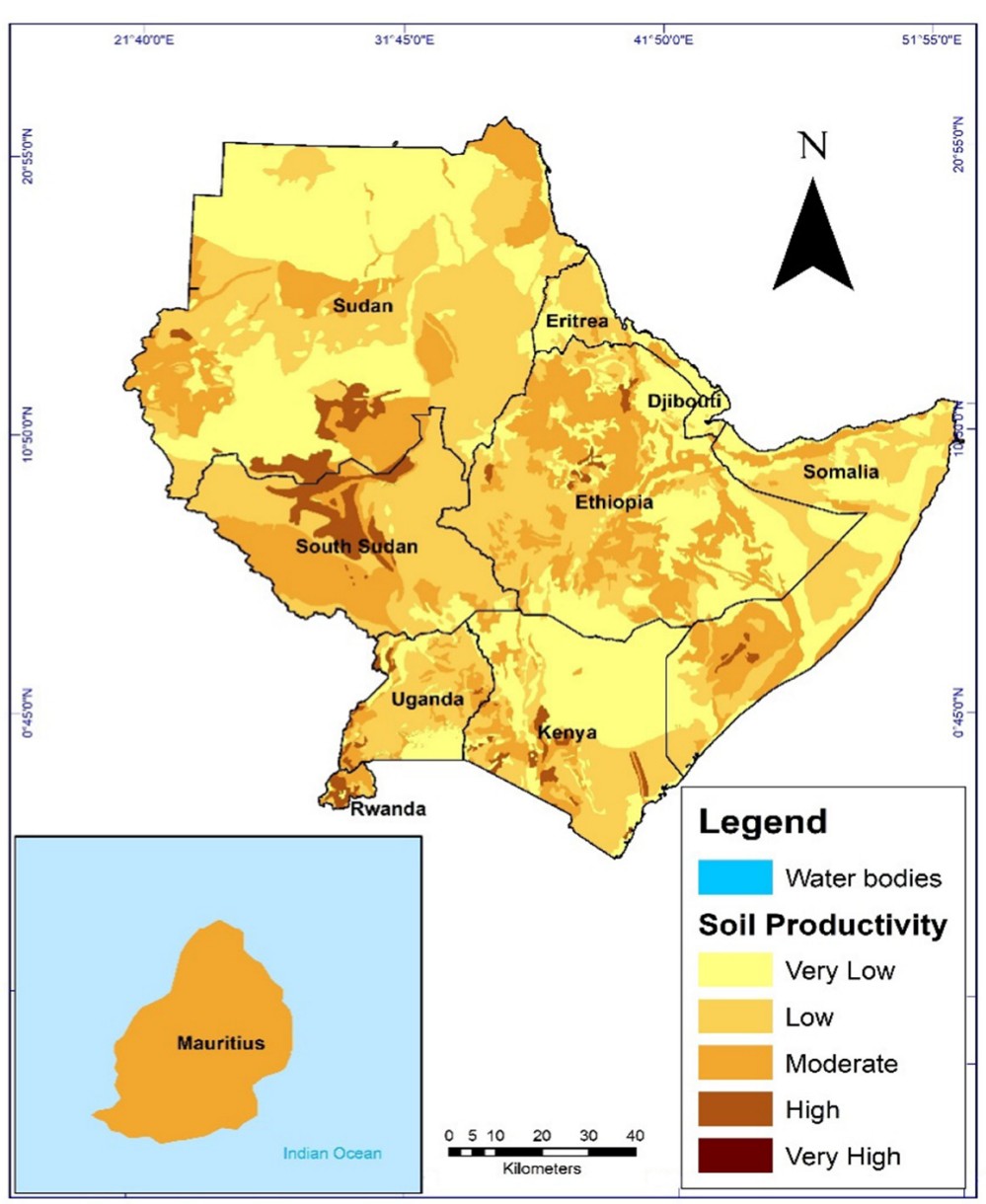

**Figure 3.** Soil Productivity status of the studied countries.

**Table 13.** Soil Productivity Index for the studied countries.

| Intensities | Djibouti | | Eritrea | | Ethiopia | | Kenya | | Mauritius | | Rwanda | | Somalia | | South Sudan | | Sudan | | Uganda | |
|---|---|---|---|---|---|---|---|---|---|---|---|---|---|---|---|---|---|---|---|---|
| | Area (km²) | % | Area (km²) | % | Area (km²) | % | Area (km²) | % | Area (km²) | % | Area (km²) | % | Area (km²) | % | Area (km²) | % | Area (km²) | % | Area (km²) | % |
| Very Low | 2290 | 10.0 | 47,213.0 | 38.4 | 244,224.4 | 21.3 | 230,020.8 | 38.8 | | | 2075.3 | 8.5 | 196,711.9 | 29.4 | 22,722.91 | 3.61 | 820,584.41 | 44.04 | 38,233.9 | 15.8 |
| Low | 19,792 | 86.2 | 65,677.1 | 53.4 | 499,035.8 | 43.6 | 268,143.8 | 45.2 | | | 4383.0 | 17.9 | 310,203.3 | 46.3 | 343,576.29 | 54.55 | 673,948.91 | 36.17 | 152,801.3 | 63.3 |
| Moderate | 587 | 2.6 | 7344.5 | 6.0 | 297,146.2 | 26.0 | 49,116.9 | 8.3 | 335.4 | 21.5 | 8300.9 | 33.8 | 75,375.4 | 11.2 | 144,531.60 | 22.95 | 131,914.60 | 7.08 | 34,397.8 | 14.3 |
| High | 281 | 1.2 | 2761.0 | 2.2 | 92,003.6 | 8.0 | 22,298.1 | 3.8 | 1225.3 | 78.5 | 995.3 | 4.1 | 82,203.0 | 12.3 | 48,413.01 | 7.69 | 177,142.46 | 9.51 | 6737.5 | 2.8 |
| Very high | | | | | 12,032.9 | 1.1 | 23,798.2 | 4.0 | | | 8797.6 | 35.8 | 5528.6 | 0.8 | 70,558.41 | 11.20 | 59,509.202 | 3.2 | 9071.0 | 3.8 |

*3.2. Characteristics of the Land Productivity Classes*

The characteristics of the eight land productivity classes are hereby presented in Table 14. The low classes are differentiated by the extent of bare patches and sparse grasslands and/or shrubs—while moderate to very high land productivity was differentiated by the extent of the standing biomass.

**Table 14.** Description of the different land productivity classes.

| Class | Description |
|---|---|
| Extremely Low | These are portions of land that are entirely bare or rocky |
| Very Low | These are portions of land that are bare to some extent |
| Low | These are portions of land that are characterised by sparse grasslands, shrubs with bare patches |
| Slightly Moderate | Portions of land with slightly moderate standing biomass (closed vegetation e.g., Closed evergreen or deciduous forest, Mosaic vegetation (grassland/shrubland/forest)/cropland) |
| Moderate | Portions of land with moderate standing biomass (closed vegetation e.g., Closed evergreen or deciduous forest, Mosaic vegetation (grassland/shrubland/forest)/cropland) |
| Slightly High | Portions of land with slightly high standing biomass (closed vegetation e.g., Closed evergreen or deciduous forest, Mosaic vegetation (grassland/shrubland/forest)/cropland) |
| High | Portions of land with high standing biomass (closed vegetation e.g., Closed evergreen or deciduous forest, Mosaic vegetation (grassland/shrubland/forest)/cropland) |
| Very High | Portions of land with very high standing biomass (closed vegetation e.g., Closed evergreen or deciduous forest, Mosaic vegetation (grassland/shrubland/forest)/cropland) |

*3.3. Land Productivity Status*

This study shows that, in the studied countries, about 10.8% of the land is of "Very high" land productivity (Table 15). About a third of the region is "Extremely low " to "low, a third is "Slightly moderate" to "Moderate", and a third is of "Slightly high" to "Very High" land productivity. At the country level, the countries with the highest spatial extent of standing biomass are Mauritius (97%), Rwanda (96%), Uganda (95%), South Sudan (89%), Ethiopia (47%) and Kenya (36%), while the least spatial extent of standing biomass is found in Djibouti (99.8%), Eritrea (69.7%), Sudan (62.1%) and Somalia (38.9%) (Table 16 and Figure 4).

**Table 15.** The regional spatial extent of land productivity in the studied countries.

| Land Productivity | Area (km$^2$) | % |
|---|---|---|
| **Extremely Low** | 71,818 | 1.3 |
| **Very Low** | 907,988 | 16.6 |
| **Low** | 900,543 | 16.4 |
| **Slightly Moderate** | 883,370 | 16.1 |
| **Moderate** | 784,025 | 14.3 |
| **Slightly High** | 1,031,753 | 18.8 |
| **High** | 310,141 | 5.7 |
| **Very High** | 592,291 | 10.8 |

**Table 16.** Country specific status of land productivity for the studied countries as per 2019.

| Intensities | Djibouti | | Eritrea | | Ethiopia | | Kenya | | Mauritius | | Rwanda | | Somalia | | South Sudan | | Sudan | | Uganda | |
|---|---|---|---|---|---|---|---|---|---|---|---|---|---|---|---|---|---|---|---|---|
| | Area (km$^2$) | % | Area (km$^2$) | % | Area (km$^2$) | % | Area (km$^2$) | % | Area (km$^2$) | % | Area (km$^2$) | % | Area (km$^2$) | % | Area (km$^2$) | % | Area (km$^2$) | % | Area (km$^2$) | % |
| Extremely Low | 4062.65 | 19.0 | 10,155 | 8.42 | 13,967 | 1.23 | 1969 | 0.3 | 2.7 | 0.25 | 409 | 1.56 | 693 | 0.11 | 99 | 0.01 | 33,609 | 1.63 | 6852 | 3.06 |
| Very Low | 16,724.80 | 78.4 | 46,544 | 38.59 | 66,545 | 5.87 | 25,838 | 4.4 | 3.2 | 0.3 | 130 | 0.49 | 25,106 | 3.97 | 198 | 0.03 | 726,185 | 35.23 | 714 | 0.32 |
| Low | 500.14 | 2.3 | 27,331 | 22.66 | 70,300 | 6.20 | 60,284 | 10.2 | 3.2 | 0.3 | 106 | 0.4 | 219,666 | 34.78 | 1148 | 0.17 | 520,585 | 25.25 | 620 | 0.28 |
| Slightly Moderate | 43.08 | 0.2 | 23,185 | 19.22 | 225,314 | 19.86 | 133,223 | 22.6 | 6.4 | 0.6 | 132 | 0.5 | 180,249 | 28.54 | 9308 | 1.39 | 311,255 | 15.1 | 655 | 0.29 |
| Moderate | 7.35 | 0.03 | 10,591 | 8.78 | 225,747 | 19.90 | 156,783 | 26.6 | 21.2 | 1.98 | 367 | 1.4 | 126,328 | 20 | 65,965 | 9.82 | 196,119 | 9.51 | 2096 | 0.94 |
| Slightly High | 2.07 | 0.010 | 2747 | 2.28 | 267,703 | 23.60 | 109,102 | 18.5 | 80.1 | 7.48 | 7446 | 28.36 | 68,041 | 10.77 | 312,420 | 46.52 | 231,528 | 11.23 | 32,684 | 14.59 |
| High | 0.05 | 0.0002 | 50 | 0.04 | 86,590 | 7.63 | 29,013 | 4.9 | 65.8 | 6.14 | 8606 | 32.78 | 7770 | 1.23 | 116,289 | 17.32 | 25,643 | 1.24 | 36,114 | 16.12 |
| Very High | 0 | 0 | 23 | 0.02 | 178,351 | 15.72 | 73,148 | 12.4 | 888.8 | 82.95 | 9060 | 34.51 | 3764 | 0.6 | 166,154 | 24.74 | 16,630 | 0.81 | 144,272 | 64.41 |

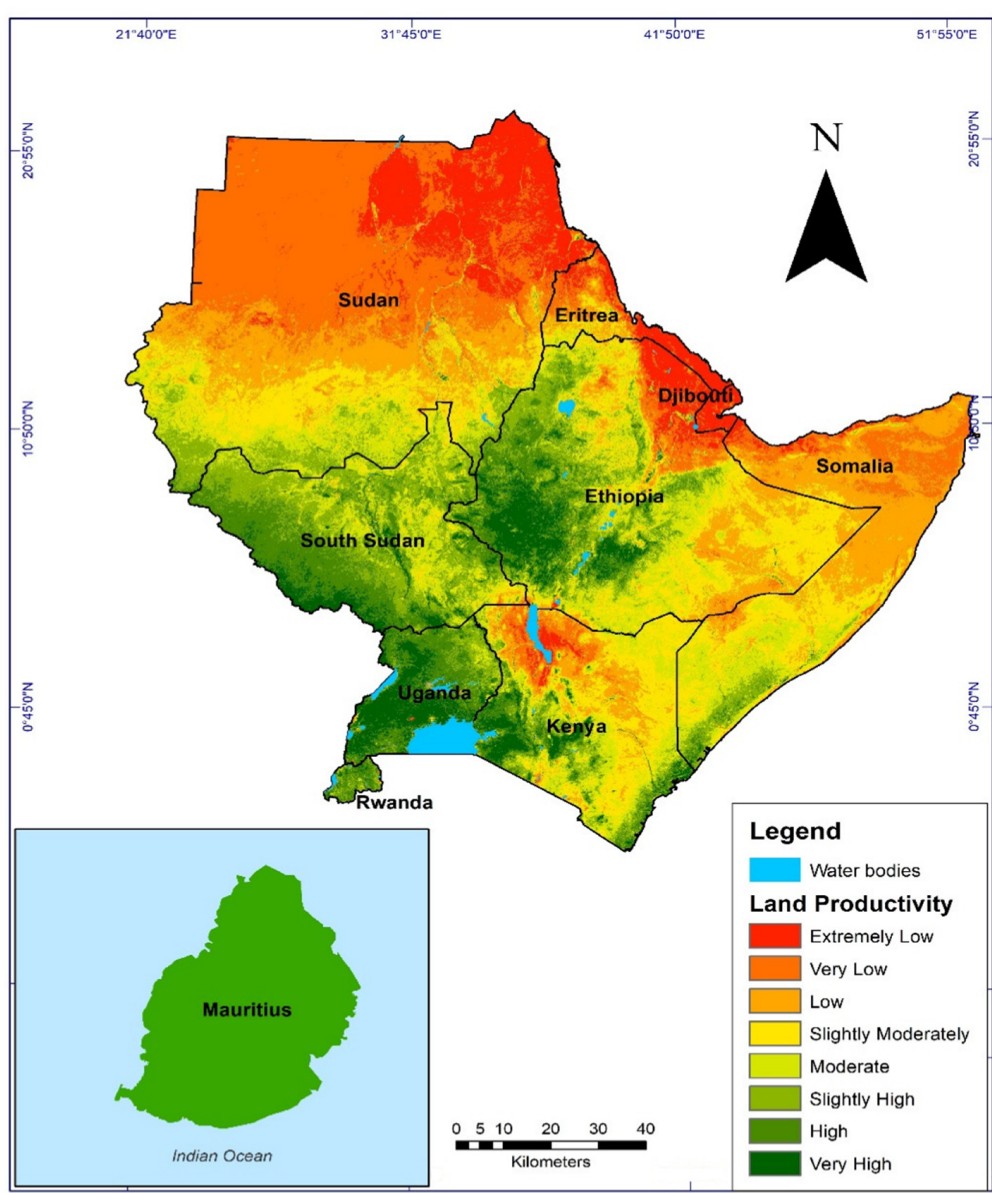

**Figure 4.** State of land productivity.

*3.4. Causes of High and Low Land Productivity Status*

As explained by the key informants, the dominant standing vegetative biomass was categorised as natural forests, woodlands, croplands, grasslands, wetlands and tree plantations. They attributed the High Land productivity status to the best soil quality and management (such as fertilization and irrigation), agricultural policies, climatic conditions and land husbandry activities (terraces, trenches, land consolidation), while they explained the low standing vegetative biomass by the high magnitude of soil erosion, floods and droughts, mining activities, urbanisation, low soil fertility, high clay content in the soils and poor farming methods, especially over-cultivation.

In addition, the determinants of soil productivity as reported by the key informants were soil moisture content, soil drainage, soil depth, soil texture/structure, soluble salt concentration, organic matter, mineral exchange capacity and overgrazing. The reported regions of high and low productivity in the ten studied countries are shown in Table 17.

**Table 17.** Land productivity status in the studied countries.

| | | Land Productivity Status | |
|---|---|---|---|
| **No** | **Country** | **Most Productive Regions** | **Least Productive Regions** |
| 1 | Uganda | West, central, east | Karamoja (Northern eastern) |
| 2 | South Sudan | Equatorial region | Bahr el Ghazal |
| 3 | Sudan | Darfur, South Kordofan and Blue Nile (& central region) | Northern region |
| 4 | Rwanda | Northern region | Southern and western |
| 5 | Somalia | Lower Jubba, Middle Jubba, Lower Shabelle, Middle Shabelle, Bay, Gado, Bakol, Hiraan, Part of Awdal, Part of north east | Nugal, Sool, East (Bari), Part of Sanaag, Part of North East (Waqooyi Galbeed) |
| 6 | Kenya | Uasin Gishu, Trans Nzoia and parts of Lake Victoria basin. Others include Muranga, Nyeri, Meru and Tharaka-Nithi (Mountain areas and western parts of Kenya) | Samburu, Kitui, Garissa, Tana River, Mandera, Turkana, Marsabit, Baringo, West Pokot, Kajiado, Kilifi, Wajir and Makueni |
| 7 | Ethiopia | Addis Ababa, Harari, Dire Dawa, Gambella, Benishangul Gumuz | Amhara, Oromia, Tigray, Somali |
| 8 | Eritrea | The western part of Eritrea (Gash Barka and some part of Anseba) and areas along the coastal zones | The Danakil area, Northern part, and central part of Eritrea |
| 9 | Djibouti | Tadjourah region, and the Mabla Mountains near Obock, Dorra, Balho | Randa, Obock, Ali Addeh, Dikhil |
| 10 | Mauritius | Northern, eastern and southern i.e., Pamplemousses, Grand Bale, Quatre Bornes, Roches Noires | Western part i.e., Noyale–Chamarel-Bel Ombre, Port Louis –Signal Mountain, Balaclava-Grand Baie–Goodlands, Roche noire-Bras D'Eau–Belle Mare-Trou D'Eau Douce-Grand Riviere Sud Est, Quatre Soeurs-Grand Sable-Bois Des Amourettes-Mahebourg-Blue Bay-Le Bouchon |

*3.5. Effect of Rainfall on Land Productivity*

The results showed a positive and strong relationship between rainfall and standing biomass in Kenya, Eritrea, Mauritius and South Sudan. The correlation is significant and moderate in Uganda and poor in Djibouti and Rwanda (Figure 5). The positive relationship implies that rainfall has a significant contribution to triggering and sustaining the availability of standing biomass compared to the countries that provided weaker correlations. As per interviewed key informants, the additional parameters which could have influenced standing biomass include topography, management and policy reforms.

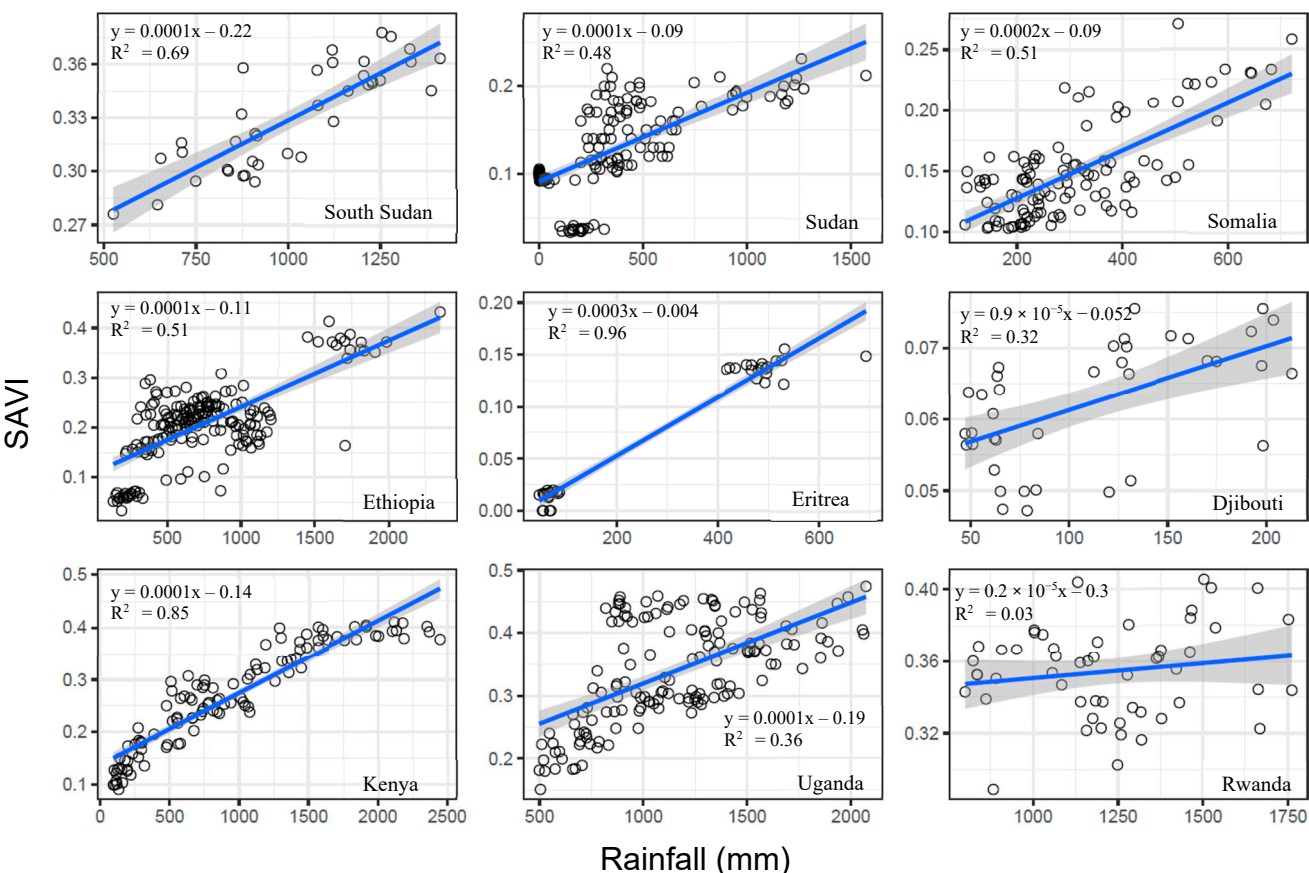

**Figure 5.** Relationship between SAVI and Rainfall (2001–2019) in the studied countries.

## 4. Discussion

### 4.1. Soil Productivity Index

This study has shown that an equal proportion of land has a "Very Low" to "Low", "Slightly moderate" to "Moderate" and" High" to "Very High" SPI. This confirms the diversity in terms of the environmental and soil-forming factors in the region [8] and the claims by Sanchez and Logan [43] that some soils of the region are acidic, infertile and often incapable of sustained agricultural production. Land productivity in Eastern Africa is highly influenced by the state of soils in their natural environment and the amount of rainfall received. Soils of high productivity are deep, permeable, with sufficient nutrient content and supply, and do not have water stress. It is worthwhile to note that most of the soils in the region have strong acidity of soluble Aluminium, which is toxic for most crop species. These include Ferralsols, Acrisols, Cambisols and Vertisols. Zake [44] noted, for example, that more than 70% of the land in Uganda is covered by Ferralsols, Vertisols and Acrisols. This is one of the reasons why about 10% of the soils in Uganda are productive. According to Msanya [45], about 52% of Tanzania is covered by Cambisols, Ferralsols and Vertisols. The SPI results are in line with Eswaran et al. [8] and report that about 16% of the land on the continent is of High soil productivity, 13% of medium and about 55% of them are unsuitable for any form of cultivated agriculture except nomadic grazing. The regional use of inorganic fertilizers, irrigation and adoption of soil and water conservation and climate-smart agriculture has remained very low.

It is worthwhile to note that, although the results are in line with previous authors' observations in the region, deviations from reality on the ground could be induced by the accuracy of the model used, the choice of factors, their relative importance and their resolutions.

### 4.2. Net Primary Productivity

Mostly the areas with "High" to "Very High" standing biomass occurred in regions with "Moderate" to "Very High" SPI and that receive adequate rainfall amount. This is in line with Lewis et al. [46] observations on the positive relationship between the Above Ground Biomass (AGB) and rainfall, clay-rich soils; C:N ratio, and soil fertility computed as the sum of base cations. Areas with high SPI which had low standing biomass were either used for cultivation or have undergone degradation. In various Eastern African countries, the private sector and individuals have been encouraged to invest in tree plantations [47]. The latter can provide affordable wood for industry and wood-based products for consumers [48]. Hence, its offset will result in the pressure on wood products from natural forests and vulnerability of forest ecosystems degradation for their conservation, protection and recreation purposes [49]. However, if well managed, they can also contribute positively to the provision of environmental and social services and livelihood support [50]. Unfortunately, the demographic pressure characterizing these countries and the region and the concomitant of social and economic development is likely to increase the demand for and consumption of wood products. Subsequently, the standing biomass will still dwindle due to deforestation for timber and wood supply, and horizontal expansion of agricultural land. For example, Bullock et al. [51] reported that in the East African region the rapid economic changes experienced in the past 30 years have been at the expense of natural ecosystems. Most East and South African nations including Kenya, Malawi, Rwanda, Tanzania, Ethiopia, Burundi, Zambia and Uganda have communities that are often involved in the conversion of woody natural habitats to less-woody cultivated or developed land cover types [16]. From 1990 to 2020, natural forest cover has decreased by 17% [52]. Deviations from realities from the ground could also be explained by the resolution of the images used in the study.

Freeman [53] also noted that the average life expectancy increased from 45 to 67 years. This change in life expectancy coupled with the population growth is likely to double the population of the region in the next 30 years. The economic changes and population growth have contributed to the growth of small urban centres in the region [54], hence encroaching on natural habitats. The increasing demands of growing urban populations on natural resources put direct and indirect pressures on natural ecosystems in the region [55]. In addition, the continuous cropping and inadequate replacement of nutrients removed in harvested materials or loss through erosion and leaching subsequently leads to soil fertility decline [56] and hence poor standing vegetation biomass. Although agroforestry has also been promoted in the region, the adoption rate has remained very low [57]. However, agroforestry is one of the key sustainable management practices which can contribute significantly to increasing standing vegetation biomass and reducing deforestation [58,59] as well as promoting biodiversity conservation.

### 4.3. Effect of Rainfall on Standing Biomass

Generally, rainfall had a significant and strong effect on the standing biomass in all the countries except Djibouti, Mauritius and Rwanda, where the correlation was very low. In Uganda, the correlation between annual rainfall and annual standing biomass was moderate. The relationship between the annual rainfall and standing biomass is explained by the type of soils, the climate and the type of tree species that grow in the various ecosystems in the different countries. The growth of trees is reduced or ceases due to limited water availability in the soils [60–62]. Water retention in the soils is a function of their depth and permeability. The majority of the soils in Eastern African countries present a good depth for water storage. For Uganda, the Karamoja region and some parts of the northern region are under semi-arid conditions and have shallow soils, respectively. The trees in the Karamoja region are dominated by savannah grassland with scattered acacia species, while the northern region is dominantly a savannah grassland region. The poor correlation between rainfall and standing biomass in Rwanda is associated with the status of land use/cover in the country. Apart from the game reserves and protected areas, most

of the country is either cultivated land or under settlements. A study by Li et al. [63] shows that, before 2000, the land-use and land cover (LULC) in Rwanda was mainly converted from forest and grassland to cropland, with the ratio being 0.72:0.28; however, after 2010, the LULC was mainly converted from forest to grassland and cropland, with the ratio of 0.83:0.17. The situation in Djibouti is explained by the low SPI and low amount of rainfall.

Therefore, much of this study was conducted in Africa, and the proposed model can be customised to assess soil and land productivity for any continent. It can also be applied from national to plot level investigations. The aim is to restore degraded landscapes or sustain vegetative biomass purposely to increase soil productivity, ecosystem services and production.

## 5. Conclusions

This study shows that, over time, the countries with the most productive soils are Mauritius, Rwanda and South Sudan. Our study further reveals that the countries with the most productive land are Mauritius, Rwanda, Uganda, South Sudan, Ethiopia and Kenya, while those with the least standing biomass are Djibouti, Eritrea, Sudan and Somalia. It also shows an association between SPI and standing biomass. The dominant productive biomes are natural forests, woodlands, croplands, grasslands, wetlands and tree plantations. Generally, there is a strong and significant positive correlation between annual rainfall and standing biomass, except in Djibouti, Uganda and Rwanda. This is favoured by soil quality, conducive climatic conditions, policies and land husbandry activities. This study demonstrates the importance of improving soil health, use of sustainable land utilisation, initiating land reforms and increasing crop-related productivity if land productivity is to be improved.

**Author Contributions:** Conceptualization, B.B. and M.J.G.M.; funding acquisition and project management, B.M. and M.N.W.; data management, M.P.; software developer, K.J.; stakeholder consultations, S.O. and W.L. All authors have read and agreed to the published version of the manuscript.

**Funding:** This project was funded by the joint initiative of the European Commission and African Union through the Regional Centre for Mapping of Resources for Development (RCMRD) under the "GMEs Africa Program" and conducted by GEO-MIK Consultants Africa on contract (CSSM/ADMIN/5/1/VI).

**Institutional Review Board Statement:** Not applicable.

**Informed Consent Statement:** Informed consent was obtained from all subjects involved in the study.

**Data Availability Statement:** Not applicable.

**Acknowledgments:** This study acknowledges the financial and technical support provided by the staff of RCMRD towards the conceptualization, validation and publication of this article. In particular, we are grateful for the support provided by Degelo Sariko Sendabo, Josephat Makanga, David Ongo Nyang'acha, Michael Ngugi Kimani, Eunice Wangui Mwangi, Esther Wambui Muigai and Joseph Murage. We also appreciate the support provided by the selected key informants from different countries contacted during the implementation of this study.

**Conflicts of Interest:** The authors declare no conflict of interest.

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
