# Peer review of "A Simplified Spatial Methodology for Assessing Land Productivity Status in Africa"

_land, doi:10.3390/land11050730_

Round 1
Reviewer 1 Report
The study is very interesting as it combines traditional land evaluation approach with the assessment of productivity indicators derived from remote sensing. Furthermore, the attempt to include degradation (erosion) assessment to the land evaluation process is also a plus.
Its findings are certainly interesting for the land science community and can be used for planning and policy purposes too.
However, there are a number of shortcomings, which need to be fixed before the paper can reach the required quality needed for publication. The ambitions of the paper are quite high and so far the condensation of the processed information is not fully successful.
While in the beginning of the paper the relationship between soil productivity and land productivity could be understood, it was confused in the later parts of the manuscript. Eg. the Modified Soil Productivity Index of equation 1. includes slope (E), which is a land attribute.
A full revision of the manuscript is needed, with clarification of soil productivity and land productivity and the related metrics.
Results of the rating system is also unclear, the presentation of the results in % is quite confusing. For example see the controversial sentences:
“In the studied countries, the productivity of land is relatively good (64%) in consideration of the very high-class intensity most times of the year.” vs “This study reveals that the soils in the studied countries are generally poor (74.2%) to support plant life on their own.”
The earlier also appears in the abstract, without clear meaning of the value.
I feel that chapter 3.5 in its current form is unnecessary. Probably a separate article in the relevant journal would better serve the aim to present this tool.
While in such large areas climate is the most important factor of productivity differences, there is little mention to this in the manuscript (except for rainfall) . Although I understand that soil is more of the interest of the authors, but this makes the research and its results quite unbalanced.
The inclusion of erosion and nutrient are good ideas, but without validation of their effect individually in a spatial context it is difficult to judge its usefulness. (Country-scale measures may be quite misleading!) Please explain in detail and provide validation.
Also, statements like “sufficient literature available of assessing land degradation in Africa [18-20]” shall be avoided, especially with providing only three references, among which there era only regional studies.
Another one is “studies that have at tempted to examine land productivity, have only concentrated at plot level, country specific and also focus on only cultivated land productivity”, which does not stand. (Please refer to AEZ and many work of Fischer et al at IIASA and FAO.)
In summary, while the research presented has the potential to be published, the manuscript needs substantial improvement before can be recommended for acceptance.
Author Response
COMMENTS REVIEW
REVIEWER 1
No |
Reviewer’s Comments |
Action taken |
Remarks |
Open Review |
|
|
|
1. |
Does the introduction provide sufficient - Must be improved |
This has been improved |
|
2. |
Is the research design appropriate? - Must be improved |
This has been improved |
|
3. |
Are the methods adequately described? - Must be improved |
This has been improved |
|
4. |
Are the results clearly presented? - Can be improved |
This has been improved |
|
5. |
Are the conclusions supported by the results? - Can be improved |
This has been improved |
|
Comments and Suggestions for Authors |
|
|
|
6 |
The study is very interesting as it combines traditional land evaluation approach with the assessment of productivity indicators derived from remote sensing. Furthermore, the attempt to include degradation (erosion) assessment to the land evaluation process is also a plus. |
|
Good to note |
7. |
However, there are a number of shortcomings, which need to be fixed before the paper can reach the required quality needed for publication. The ambitions of the paper are quite high and so far the condensation of the processed information is not fully successful. |
These short comings have been addressed in all the sections of the paper |
|
8. |
While in the beginning of the paper the relationship between soil productivity and land productivity could be understood, it was confused in the later parts of the manuscript. Eg. the Modified Soil Productivity Index of equation 1. includes slope (E), which is a land attribute. |
The explanation has been streamlined and now it is clearer |
|
9. |
A full revision of the manuscript is needed, with clarification of soil productivity and land productivity and the related metrics. |
A revision has been made from introduction to discussion |
|
10. |
Results of the rating system is also unclear, the presentation of the results in % is quite confusing. For example see the controversial sentences: “In the studied countries, the productivity of land is relatively good (64%) in consideration of the very high-class intensity most times of the year.” vs “This study reveals that the soils in the studied countries are generally poor (74.2%) to support plant life on their own.” The earlier also appears in the abstract, without clear meaning of the value. |
The confusion in statistics has been streamed |
|
11 |
I feel that chapter 3.5 in its current form is unnecessary. Probably a separate article in the relevant journal would better serve the aim to present this tool. |
This part has been removed as recommended |
|
12 |
While in such large areas climate is the most important factor of productivity differences, there is little mention to this in the manuscript (except for rainfall) . Although I understand that soil is more of the interest of the authors, but this makes the research and its results quite unbalanced. |
|
The proposed methodology was developed in consideration of rainfall and soil as one of the key parameters |
13 |
The inclusion of erosion and nutrient are good ideas, but without validation of their effect individually in a spatial context it is difficult to judge its usefulness. (Country-scale measures may be quite misleading!) Please explain in detail and provide validation. |
Soil erosion estimation is good because it is an intermediary of soil and land productivity. The products were validated through stakeholder consultation |
|
14 |
Also, statements like “sufficient literature available of assessing land degradation in Africa [18-20]” shall be avoided, especially with providing only three references, among which there era only regional studies. Another one is “studies that have at tempted to examine land productivity, have only concentrated at plot level, country specific and also focus on only cultivated land productivity”, which does not stand. (Please refer to AEZ and many work of Fischer et al at IIASA and FAO.) |
These statements have been modified |
|
15 |
In summary, while the research presented has the potential to be published, the manuscript needs substantial improvement before can be recommended for acceptance. |
|
The manuscript has been enormously improved |
Reviewer 2 Report
I think that stady is a very interesting work of You, but it should be improved in many way. First of all, there are some general statement is the Abstract, whiches are do not conclude from the results of Your work. Also there are some mistakes in the other subdivisions. I marked that unceraibs in the text. I also took some remarks and questions in the text.

Author Response
REVIEWER 2 |
|||
Open Review |
|
|
|
Are the methods adequately described? Can be improved |
This has been improved |
|
|
Are the results clearly presented? Can be improved |
This has been improved |
|
|
Are the conclusions supported by the results? Can be improved |
This has been improved |
|
|
English language and style Moderate English changes required |
This has been improved |
|
|
Comments and Suggestions for Authors |
|
|
|
I think that study is a very interesting work of You, but it should be improved in many ways. First, there are some general statement is the Abstract, which are do not conclude from the results of Your work. In addition, there are some mistakes in the other subdivisions. I marked that unceraibs in the text. I also took some remarks and questions in the text. |
The abstract has been modified |
|
|
These are too general statements and do not derive from the results of the study |
The paragraph has been modified |
Much as the statement provided was from results, it has been modified |
|
Please show the climate and soil conditions of the area, because these factors play important role in soil and land productivity. Also it would be useful some facts about vegetation, produced agricultural crops. |
This information has been added as proposed |
|
|
Reviewer 3 Report
Interesting and important topic but not a scientific study. The study does not compile to the criteria of repeatability, i.e. the reader (an informed colleague) should be able to repeat the study based the information in the methodology section. It is not currently possible.
The "validation" of reported quantities on soil and land productivity is purely qualitative and not useful.
No analysis of uncertainty is performed.
There is no critical evaluation of the input data used. How old/valid is the FAO data set for example?
Use of units etc. are sloppy and sometimes missing.
From a remote sensing perspective would perhaps the MOD17 product be a better alternative to use instead of SAVI.
A major rework is needed before publication.
Author Response
REVIEWER 3 |
|||
|
Open Review |
|
|
1 |
Extensive editing of English language and style required |
This has been improved |
|
2 |
Does the introduction provide sufficient background and include all relevant references? Must be improved |
This has been improved |
|
3 |
Is the research design appropriate? Must be improved |
This has been improved |
|
4 |
Are the methods adequately described? Must be improved |
This has been improved |
|
5 |
Are the results clearly presented? Must be improved |
This has been improved |
|
6 |
Are the conclusions supported by the results? Must be improved |
This has been improved |
|
|
Comments and Suggestions for Authors |
|
|
7 |
Interesting and important topic but not a scientific study. The study does not compile to the criteria of repeatability, i.e. the reader (an informed colleague) should be able to repeat the study based the information in the methodology section. It is not currently possible. |
The methodology has been made more clearer for repeatability |
|
8 |
The "validation" of reported quantities on soil and land productivity is purely qualitative and not useful. |
The idea was to simplify the steps of producing land productivity maps |
|
9 |
No analysis of uncertainty is performed |
This was not part of the study objectives |
|
10 |
There is no critical evaluation of the input data used. How old/valid is the FAO data set for example? |
The datasets were collated with country specific datasets |
|
11 |
Use of units etc. are sloppy and sometimes missing |
|
|
12 |
From a remote sensing perspective would perhaps the MOD17 product be a better alternative to use instead of SAVI. |
SAVI was a better option |
|
13 |
A major rework is needed before publication |
The manuscript has been overhauled |
|
Round 2
Reviewer 1 Report
In some parts you talk about the level of productivity, in other parts about the percentage arae cover of productive land and it is still not always clear which is discussed in the actual part. Please make it more clear.
Overall I see the improvement of the paper and recommend it for publication.
Author Response
In some parts you talk about the level of productivity, in other parts about the percentage arae cover of productive land and it is still not always clear which is discussed in the actual part. Please make it more clear.
The parts with the level of productivity and percentage area cover of productivity have been modified to read well. It is now more clearer